# Differentially Private n-gram Extraction[*]

**Kunho Kim**
Microsoft
kuki@microsoft.com

**Sivakanth Gopi**
Microsoft Research
sigopi@microsoft.com

**Janardhan Kulkarni**
Microsoft Research
jakul@microsoft.com

**Sergey Yekhanin**
Microsoft Research
yekhanin@microsoft.com

## Abstract

We revisit the problem of $n$-gram extraction in the differential privacy setting. In this problem, given a corpus of private text data, the goal is to release as many $n$-grams as possible while preserving user level privacy. Extracting $n$-grams is a fundamental subroutine in many NLP applications such as sentence completion, response generation for emails etc. The problem also arises in other applications such as *sequence mining*, and is a generalization of recently studied differentially private set union (DPSU). In this paper, we develop a new differentially private algorithm for this problem which, in our experiments, significantly outperforms the state-of-the-art. Our improvements stem from combining recent advances in DPSU, privacy accounting, and new heuristics for pruning in the tree-based approach initiated by Chen et al. (2012) [CAC12].

## 1   Introduction

We revisit the problem of $n$-gram extraction in the differential privacy setting. In this problem, we are given a set of $N$ users, and each user has some text data, which can be a collection of emails, documents, or conversation history. An $n$-gram is any sequence of $n$ consecutive words that appears in the text associated with some user. For example, suppose there are two users $u_1$ and $u_2$, and they have texts "Serena Williams is a great tennis player" and "Erwin Schrodinger wrote a book called What is Life". Then, 'great tennis player' is a 3-gram as it appears in the text of $u_1$, 'book called What is Life' is a valid 5-gram as it appears in the text of the $u_2$. On the other hand, 'tennis player Serena' is not a 3-gram as that sequence does not appear in the text of either of the users. Similarly 'wrote called What' is not a 3-gram as it does not appear as a contiguous subsequence of either text. Our goal is to extract as many $n$-grams[2] as possible, of all different lengths up to some maximum length $T$, while guaranteeing differential privacy.

Our motivation to study this question comes from its applications to Natural Language Processing (NLP) problems. The applications such as suggested replies for e-mails and dialog systems rely on the discovery of $n$-grams, and then training a DNN model to rank them [HLLC14, KKR$^+$16, CLB$^+$19, DBS19]. However, $n$-grams used for training come from individuals, and may contain sensitive information such as social security numbers, medical history, etc. Users may be left vulnerable if personal information is revealed (inadvertently) by an NLP model. For example, a model could complete a sentence or predict the next word that can potentially reveal personal information of the

---

[*]Code available at https://github.com/microsoft/differentially-private-ngram-extraction
[2]In some papers, an $n$-gram model specifically refers to a probabilistic prediction model based on Markov chains. We do not make any probabilistic assumptions on how user data is generated.

users in the training set [CLE$^+$19]. Therefore, algorithms that allow the public release of the $n$-grams while preserving privacy are important for NLP models that are trained on the sensitive data of users.

In this paper we study private $n$-gram extraction problem using the rigorous notion of differential privacy (DP), introduced in the seminal work of Dwork et al. [DMNS06].

**Definition 1.1** (Differential Privacy [DR14]). A randomized algorithm $\mathcal{A}$ is $(\varepsilon,\delta)$-differentially private if for any two neighboring databases $D$ and $D'$, which differ in exactly the data pertaining to a single user, and for all sets $\mathcal{S}$ of possible outputs:

$$\Pr[\mathcal{A}(D) \in \mathcal{S}] \le e^{\varepsilon} \Pr[\mathcal{A}(D') \in \mathcal{S}] + \delta.$$

We consider the $n$-gram extraction problem guaranteeing user level differential privacy. We formalize the problem as follows. Let $\Sigma$ be some vocabulary set of words. Define $\Sigma^k = \Sigma \times \Sigma \times \cdots \times \Sigma$ ($k$ times) to be the set of length $k$ sequences of words from $\Sigma$, elements of $\Sigma^k$ are called $k$-grams. Let $\Sigma^* = \cup_{k \ge 0} \Sigma^k$ denote arbitrary length sequences of words from $\Sigma$, an element $w \in \Sigma^*$ is called *text*. If $w = a_1 a_2 \ldots a_m$ where $a_i \in \Sigma$, any length $k$ *contiguous subsequence* $a_i a_{i+1} \ldots a_{i+k-1}$ of $w$ is called a $k$-gram present in $w$. The set of all $k$-grams inside a text $w$ are denoted by $G_k(w)$ and we denote by $G(w) = \cup_k G_k(w)$ the set of all $n$-grams of all lengths inside $w$.

**Problem 1.1** (DP $n$-gram Extraction (DPNE)). Let $\Sigma$ be some vocabulary set, possibly of unbounded size and let $T$ be the maximum length of $n$-grams we want to extract. Suppose we are given a database $D$ of users where each user $i$ has some text $w_i \in \Sigma^*$. Two such databases are adjacent if they differ in exactly 1 user. We want an $(\varepsilon,\delta)$-differentially private algorithm $A$ which outputs subsets $S_1, S_2, \ldots, S_T$, where $S_k \subset \Sigma^k$, such that the size of each $S_k$ is as large as possible and $S_k \setminus \cup_i G_k(w_i)$ is as small as possible.

Note that in our formulation, we assign the same weight to $n$-grams irrespective of their length; that is, both a 8-gram and a 2-gram carry equal weight of 1. While one can study weighted generalizations of the DPNE problems, both from an algorithm design perspective and the application of DPNE to NLP problems, our formulation captures the main technical hurdles in this space.

Many variants of $n$-gram discovery problems, closely related to DPNE, have been studied in the literature [CAC12, XSC$^+$15, XCS$^+$16, WXY$^+$18]. [CAC12] study this problem assuming a certain probabilistic Markov chain model of generating $n$-grams. [XSC$^+$15, XCS$^+$16] study the problem of mining frequent sequences. Another set of problems closely related to DPNE are mining or synthesizing trajectory data; see [HCM$^+$15, CFDS12] and references there in. While we build upon some of the ideas in these works, to the best of our knowledge, the specific version of $n$-gram extraction problem formalized in DPNE has not been studied before.

A special case of DPNE problem, recently introduced by Gopi et al. [GGK$^+$20], is called Differentially Private Set Union (DPSU). In this problem, we are given a possibly unbounded universe of elements, and each user holds a subset of these elements. The goal is to release the largest possible subset of the union of elements held by the users in a differentially private way. This problem can be considered simply as extracting 1-grams. Another way to relate the problems is to assume that every possible $n$-gram as a separate item in the DPSU problem. While these interpretations do imply that one can use the algorithms designed for DPSU to solve DPNE, the algorithms for DPSU fail to exploit the inherent structure of our new problem. In particular, note that if an algorithm for DPNE releases an $n$-gram of size 8, then one could extract *all* possible subgrams without any privacy cost by simply performing a post processing operation on the output of the algorithm. This structure is at the heart of the DPNE problem, and algorithms for DPSU do not take into account this. Not surprisingly, they do not give good utility as demonstrated in our experiments.

**Our Contributions**  In this work we design new algorithms for the DPNE problem. The main contributions of the work are:

- By combining ideas from the recent DPSU work with the tree based approach of [CAC12], we develop new differentially private algorithms to solve the DPNE problem. We also show an efficient implementation of our algorithm via an implicit histogram construction. Moreover, our algorithms can be easily implemented in the MAP-REDUCE framework, which is an important consideration in real-world systems.

- Using a Reddit dataset, we show that our algorithms significantly improve the size of output $n$-grams compared to direct application of DPSU algorithms. Our experiments show

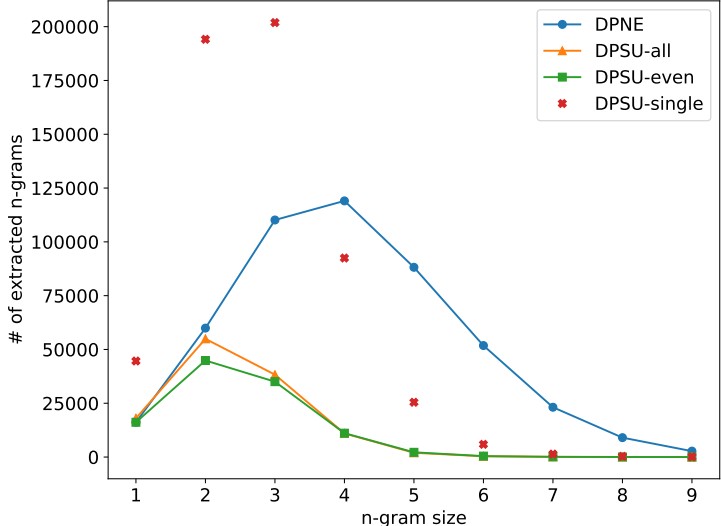

Figure 1: The figure illustrates the performance of DPNE algorithm compared to various ways one can apply the DPSU algorithm for $n$-gram extraction. Here 'DPSU-all' refers to running DPSU on all the different length $n$-grams together. 'DPSU-even' refers to splitting the privacy budget evenly and running DPSU to learn $k$-grams separately for each $k$. 'DPSU-single' refers to spending all the privacy budget to learn $k$-grams for a single $k$. Note that for large $k$, DPNE learns many more $k$-grams than DPSU even when DPSU uses all its privacy budget to learn just $k$-grams for that particular value of $k$. Here $\varepsilon = 4, \delta = 10^{-7}$.

that DPNE algorithms extract more longer $n$-grams compared to DPSU even if the DPSU algorithm spent all its privacy budget on extracting $n$-grams of a particular size (see Figure 1). Moreover, we recover most of the long $n$-grams which are used by at least 100 users in the database (see Figure 4).

Our algorithms have been used in industry to make a basic subroutine in an NLP application differentially private.

## 2   An Algorithm for DPNE

In this section we describe our algorithm for DPNE. The pseudocode is presented in Algorithm 1. The algorithm iteratively extracts $k$-grams for $k = 1, 2, \ldots, T$, i.e., the algorithm uses the already extracted $(k-1)$-grams to extract $k$-grams. Let $S_k$ denote the extracted set of $k$-grams. The main features of the algorithm are explained below.

**Build a vocabulary set (1-grams) using DPSU:**   As the first step, we use the DPSU algorithm from [GGK+20] to build a vocabulary set (1-grams), i.e., $S_1$. One can use any update policy from [GGK+20] in the DPSU algorithm. We use the weighted gaussian policy because it is simple and it scales well for large datasets. The DPSU algorithm with weighted gaussian update policy is presented in the Appendix B (Algorithm 5) for reference.

**DPSU-like update policy in each iteration:**   In each iteration, we build a histogram $H_k$ on $k$-grams using the weighted gaussian update policy from [GGK+20].[3] If we run DPSU as a blackbox again to learn $k$-grams (not making use of $S_{k-1}$), then algorithm will perform very poorly. This is because the number of unique $k$-grams grows exponentially with $k$. This causes the user budget to get spread too thin among the exponentially many $k$-grams. Therefore very few $k$-grams will get

---

[3]One can use any update policy from [GGK+20], we use weighted gaussian because of its simplicity and that it scales well to large datasets. The noise added will depend on the particular update policy chosen.

**Algorithm 1:** Algorithm for differentially private $n$-gram extraction

---

**Input:** A set of $N$ users where each user $i$ has some text $w_i$. $T$: maximum length of ngrams to be extracted

$\Delta_1, \Delta_2, \ldots, \Delta_T$: maximum contribution parameters

$\rho_1, \rho_2, \ldots, \rho_T$: Threshold parameters

$\sigma_1, \sigma_2, \ldots, \sigma_T$: Noise parameters.

**Output:** $S_1, S_2, \ldots, S_T$ where $S_k$ is a set of $k$-grams

`// Run DPSU to learn 1-grams`

$S_1 \leftarrow$ Run DPSU with weighted gaussian update policy using $\Delta_1, \rho_1, \sigma_1$ to get a set of 1-grams ;

$V_1 \leftarrow S_1$;

`// Iteratively learn `$k$`-grams`

**for** $k = 2$ **to** $T$ **do**

    $V_k \leftarrow (S_1 \times S_{k-1}) \cap (S_{k-1} \times S_1)$ ;        `// Calculate valid `$k$`-grams`

    `// Add all the valid `$k$`-grams with weight 0 to a histogram `$H_k$

    **for** $u$ *in* $V_k$ **do**

        $H_k[u] \leftarrow 0$;

    `// Build a weighted histogram using `weighted gaussian` policy`

    **for** $i = 1$ **to** $N$ **do**

        $W_i^k \leftarrow G_k(w_i)$ ;        `// Set of `$k$`-grams in text `$w_i$

        $U_i \leftarrow W_i^k \cap V_k$ ;        `// Prune away invalid `$k$`-grams`

        `// Limit user contributions`

        **if** $|U_i| > \Delta_k$ **then**

            $U_i \leftarrow$ Randomly choose $\Delta_k$ items from $U_i$;

        **for** $u$ *in* $U_i$ **do**

            $H_k[u] \leftarrow H_k[u] + \frac{1}{\sqrt{|U_i|}}$;

    `// Add noise to `$H_k$` and output `$k$`-grams which cross the threshold `$\rho_k$

    $S_k = \{\}$ (empty set);

    **for** $u \in H_k$ **do**

        **if** $H_k[u] + N(0, \sigma_k^2) > \rho_k$ **then**

            $S_k \leftarrow S_k \cup \{u\}$;

Output $S_1, S_2, \ldots, S_T$;

---

enough weight to get output by the DPSU algorithm. To avoid this we will *prune* the search space for $S_k$.

**Pruning using valid $k$-grams:** Let us imagine that $S_k$ is the set of "popular" $k$-grams, i.e., it occurs in the text of many users. Then for a $k$-gram to be popular, both the $(k-1)$-grams inside it have to be popular. If we approximate the popular $(k-1)$-grams with $S_{k-1}$, the set of extracted $(k-1)$-grams, then we can narrow down the search space to $V_k = (S_{k-1} \times S_1) \cap (S_1 \times S_{k-1})$. This set $V_k$ is called the set of valid $k$-grams. Therefore when we build the histogram $H_k$ to extract $k$-grams, we will throw away any $k$-grams which do not belong to $V_k$ (this is the *pruning* step). Pruning significantly improves the performance of the algorithm as shown in the experiments (see Section 3). Pruning also results in the following nice property for the output of Algorithm 1.

**Proposition 2.1.** The output of Algorithm 1, $S_1 \cup S_2 \cup \cdots \cup S_T$, is downward closed w.r.t taking subgrams. In particular, we cannot improve the output of the algorithm by adding subgrams of the output to itself.

*Proof.* We will prove that any collection of ngrams $S = S_1 \cup S_2 \cup \cdots \cup S_T$ (where $S_k$ are $k$-grams) is downward closed iff $S_k \subset (S_1 \times S_{k-1}) \cap (S_{k-1} \times S_1)$ for all $k$. One direction is obvious, if $S$ is downward closed then $S_k \subset (S_1 \times S_{k-1}) \cup (S_{k-1} \times S_1)$ because if $a_1 a_2 \ldots a_k \in S$, then $a_1 a_2 \ldots a_{k-1}, a_2 a_3 \ldots a_k \in S_{k-1}$ and $a_1, a_2, \ldots, a_k \in S_1$. The other direction can be proved by induction on $T$. Suppose $S_k \subset (S_1 \times S_{k-1}) \cap (S_{k-1} \times S_1)$ for every $k$. By induction

$S_1 \cup S_2 \cup \cdots \cup S_{T-1}$ is downward closed. If $w = a_1 a_2 \ldots a_T \in S_T$, then any subgram of $w$ is either a subgram of $a_1 a_2 \ldots a_{T-1} \in S_{T-1}$ or a subgram of $a_2 a_3 \ldots a_T \in S_{T-1}$. By induction, that subgram has to lie in $S_1 \cup S_2 \cup \cdots \cup S_{T-1}$. $\qquad\square$

**Controlling spurious $n$-grams using $\rho_k$:** In our privacy analysis (Section 2.1), we will show that the privacy of the DPNE algorithm depends only on $\rho_1$ and $\sigma_1, \sigma_2, \ldots, \sigma_T$. In particular, $\rho_2, \ldots, \rho_T$ do not affect privacy. Instead, they are used to control the number of *spurious* ngrams that we extract, i.e., ngrams which are not actually used by any user but output by the algorithm.

**Proposition 2.2.** For $k \geq 2$, the expected number of spurious $k$-grams output by Algorithm 1 is at most $|V_k|(1 - \Phi(\rho_k/\sigma_k))$ where $\Phi$ is the Gaussian CDF. And the algorithm will not output any spurious 1-grams.

*Proof.* A spurious $k$-gram will have zero weight in the histogram $H_k$ that the algorithm builds. So after adding $N(0, \sigma_k^2)$ noise, the probability that it will cross the threshold $\rho_k$ is exactly $1 - \Phi(\rho_k/\sigma_k)$. $\qquad\square$

Larger we set $\rho_k$, smaller the number of spurious $k$-grams. But setting large $\rho_k$ will reduce the number of non-spurious $k$-grams extracted by the algorithm. So $\rho_2, \ldots, \rho_T$ should be set delicately to balance this tension. One convenient choice of $\rho_k$ for $k \geq 2$ is to set,

$$\rho_k = \sigma_k \Phi^{-1} \left( 1 - \eta \min \left\{ 1, \frac{|S_{k-1}|}{|V_k|} \right\} \right)$$

for some $\eta \in (0, 1)$. This implies that the expected number of spurious $k$-grams output is at most $\eta \min\{|S_{k-1}|, |V_k|\}$ by Proposition 2.2. And the total number of spurious ngrams output is at most $\eta(|S_1| + |S_2| + \cdots + |S_{T-1}|)$. Therefore spurious ngrams output by the algorithm are at most an $\eta$-fraction of all the ngrams output.

**Scaling up DPNE:** For extremely large datasets, Algorithm 1 can become impractical to run. The main difficulty is that the set of valid $k$-grams $V_k = (S_1 \times S_{k-1}) \cap (S_{k-1} \times S_1)$ is hard to compute explicitly if $|S_1|$ and $|S_{k-1}|$ are both extremely large. Luckily, we can easily modify the DPNE algorithm to make it scalable. The trick is to never actually compute the set of valid $k$-grams explicitly. Observe it is trivial to check if a given $k$-gram is valid or not (asumming we know $S_1$ and $S_{k-1}$). Thus we implement the pruning step implicitly without computing $V_k$. The next problem is building the histogram $H_k$ on $V_k$. Note that any spurious $k$-gram will have weight 0 in $H_k$ after all the users update it. So instead of explicitly inserting the spurious $k$-grams into $H_k$ with weight 0, we implicitly assume that they are present. When we add noise to the histogram and output all the $k$-grams which cross the threshold $\rho_k$, the number of spurious $k$-grams that should have been output follow the binomial distribution $B_k \sim \text{Binomial}(|V_k| - |\text{supp}(H_k)|, \Phi(-\rho_k/\sigma_k))$. So we can sample $B_k$ spurious $k$-grams from $V_k \setminus \text{supp}(H_k)$, then we can just add them to the output set $S_k$ at the end. And generating a random sample from $V_k \setminus \text{supp}(H_k)$ is easy. Sample a random element from $w \in S_{k-1} \times S_1$ and output if $w \in (S_{k-1} \times S_1) \setminus \text{supp}(H_k)$, else repeat. Combining these ideas we can implement a scalable version of DPNE which is included in Appendix A (Algorithm 1). Our experiments use this faster and scalable version of DPNE.

## 2.1 Privacy Analysis

To analyse the privacy guarantees of the DPNE algorithm (Algorithm 1), we will need a few preliminaries.

**Proposition 2.3** ([DRS19]). The composition of Gaussian mechanisms with $\ell_2$-sensitivity 1 and noise parameters $\sigma_1, \sigma_2, \ldots, \sigma_T$ has the same privacy as a Gaussian mechanism with $\ell_2$-sensitivity 1 and noise parameter $\sigma$ where:

$$\frac{1}{\sigma^2} = \sum_{i=1}^{T} \frac{1}{\sigma_i^2}.$$

**Proposition 2.4** ([BW18]). For any $\varepsilon > 0$, the Gaussian mechanism with $\ell_2$-sensitivity 1 and noise parameter $\sigma$ satisfies $(\varepsilon, \delta)$-DP where

$$\delta = \Phi \left( -\varepsilon \sigma + \frac{1}{2\sigma} \right) - e^{\varepsilon} \cdot \Phi \left( -\varepsilon \sigma - \frac{1}{2\sigma} \right).$$

We are now ready to prove the privacy of our DPNE algorithm.

**Theorem 2.1.** Let $\varepsilon > 0$ and $0 < \delta < 1$. Let $\sigma^*$ be obtained by solving the equation

$$\frac{\delta}{2} = \Phi\left(-\varepsilon\sigma^* + \frac{1}{2\sigma^*}\right) - e^\varepsilon \cdot \Phi\left(-\varepsilon\sigma^* - \frac{1}{2\sigma^*}\right) \tag{1}$$

where $\Phi$ is the CDF of a standard Gaussian. Then Algorithm 1 is $(\varepsilon, \delta)$-DP if we set $\rho_1, \sigma_1, \ldots, \sigma_T$ as follows:

$$\frac{1}{\sigma^*} = \sqrt{\frac{1}{\sigma_1^2} + \frac{1}{\sigma_2^2} + \cdots + \frac{1}{\sigma_T^2}},$$

$$\rho_1 = \max_{1 \leq t \leq \Delta_1}\left(\frac{1}{\sqrt{t}} + \sigma_1 \Phi^{-1}\left(\left(1 - \frac{\delta}{2}\right)^{1/t}\right)\right).$$

*Proof.* The privacy of DPSU algorithm (Algorithm 5) is given by the composition of a Gaussian mechanism with $\ell_2$-sensitivity 1 and noise $\sigma_1$ composed with $(0, \delta/2)$-algorithm as shown in [GGK+20] if we set $\rho_1$ as shown. The construction of $k$-grams is a Gaussian mechanism with $\ell_2$-sensitivity 1 and noise $\sigma_k$. By Proposition 2.3, the composition of all these mechanisms is the composition of a Gaussian mechanism with noise $\sigma^*$ and a $(0, \delta/2)$-mechanism. Now applying Proposition 2.4 and using the simple composition theorem for DP and completes the proof.[4] □

## 3 Experiments

In this section, we empirically evaluate the performance of our algorithms on two datasets: Reddit and MSNBC. The Reddit data set is a natural language dataset used extensively in NLP applications, and is taken from TensorFlow repository.[5] The MSNBC dataset consists page visits of users who browsed msnbc.com on September 28, 1999, and is recorded at the level of URL and ordered by time.[6] This dataset has been used in the literature in frequent sequence mining problems, which is a closely related problem to ours. Tables 1, 2 summarize the salient properties of these datasets. As primary focus of our paper is on NLP applications, we perform more extensive experiments on Reddit dataset, which also happens to be a significantly larger dataset compared to the MSNBC dataset.

Table 1: Dataset Statistics

|        | # Users   | # Posts   | # Sequences/Users | # Sequences | # Unique Sequences |
|--------|-----------|-----------|-------------------|-------------|--------------------|
| Reddit | 1,217,516 | 3,843,330 | 3653.81           | 43.39B      | 23.24B             |
| MSNBC  | 989,818   | 989,818   | 18.02             | 17.84M      | 357.84K            |

Table 2: Average number of sequences per user calculated for each sequence length

|        | 1      | 2      | 3      | 4      | 5      | 6      | 7      | 8      | 9      | Total   |
|--------|--------|--------|--------|--------|--------|--------|--------|--------|--------|---------|
| Reddit | 497.60 | 470.76 | 444.37 | 418.77 | 393.72 | 369.32 | 345.65 | 322.80 | 300.82 | 3563.81 |
| MSNBC  | 4.67   | 3.67   | 3.04   | 2.57   | 2.19   | 1.88   |        |        |        | 18.02   |

### 3.1 Experiments on Reddit Dataset

Throughout this section we fix $T = 9, \varepsilon = 4, \delta = 10^{-7}, \Delta_1 = \cdots = \Delta_9 = \Delta_0 = 300, \eta = 0.01$ unless otherwise specified.

#### 3.1.1 Hyperparameter tuning

There are several parameters in Algorithm 1 such as $\sigma_k, \Delta_k, \rho_k$. We study the effect of important hyperparameters on the number of $n$-grams extracted by our algorithm.

---

[4]The simple composition theorem states that if $M_i$ satisfies $(\varepsilon_i, \delta_i)$-DP for $i = 1, 2$, then the composition of $M_1, M_2$ satisfies $(\varepsilon_1 + \varepsilon_2, \delta_1 + \delta_2)$-DP.

[5]https://www.tensorflow.org/datasets/catalog/reddit

[6]https://archive.ics.uci.edu/ml/datasets/msnbc.com+anonymous+web+data

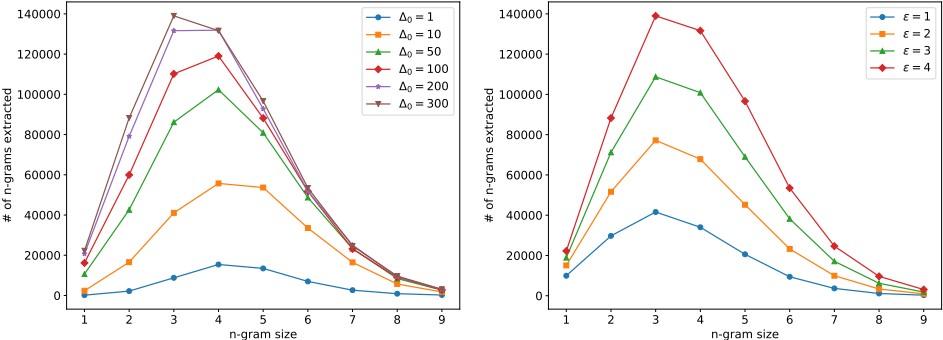

Figure 2: The figures illustrate the effect of hyperparameters on DPNE algorithm: User contribution $\Delta_0$ (left), privacy parameter $\varepsilon$ (right).

**Setting $\Delta_k$ and $\varepsilon$:** This is quite similar to setting $\Delta_0$ parameter in the DPSU algorithm from [GGK+20]. A good starting point is to set $\Delta_k$ around the median number of valid $k$-grams that users have in their text. In our experiments we set $\Delta_1 = \Delta_2 = \cdots = \Delta_k = \Delta_0$. The effect of $\Delta_0$ on the performance of the algorithm is shown in Figure 2. As predicted the performance of the algorithm improves with increasing $\Delta_0$ until $\Delta_0 \approx 300$ which is approximately the average number of $k$-grams per user in the Reddit dataset (see Table 3). Also the right side of Figure 2 shows the effect of varying $\varepsilon$. As expected, we can extract more $n$-grams for all lengths by increasing $\varepsilon$, which leads a weaker differential privacy guarantee.

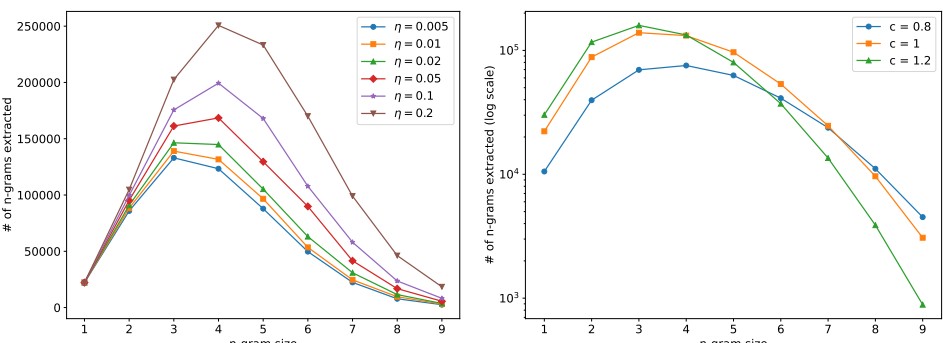

Figure 3: The figures illustrate the effect of hyperparameters on DPNE algorithm: Fraction of spurious $n$-grams $\eta$ (left), and privacy budgeting parameter $c$ (we set $\sigma_k = c\sigma_{k-1}$) (right).

**Setting $\rho_k$ and $\eta$:** We have already discussed how to set $\rho_k$. These should be set based on the fraction $\eta$ of the spurious $k$-grams we are willing to tolerate in the output. The effect of $\eta$ on the performance of the algorithm is shown in Figure 3. As expected, increasing $\eta$ increases the number of extracted $n$-grams for all lengths. Table 3 shows the actual ratio of spurious n-grams extracted by varying $\eta$ value, which experimentally confirms that the algorithm extracts at most $\eta$-fraction of the output.

Table 3: Actual ratio of spurious n-grams extracted by varying $\eta$

| $\eta$ | 0.005 | 0.01 | 0.02 | 0.05 | 0.1 | 0.2 |
|---|---|---|---|---|---|---|
| Actual Ratio | 0.004 | 0.088 | 0.017 | 0.419 | 0.086 | 0.168 |

**Setting $\sigma_k$:** As can be observed, each iteration of $k$-gram extraction for $k = 1, 2, \ldots, T$ consumes some privacy budget. If we set a small value of $\sigma_k$, we will consume more privacy budget constructing $k$-grams. Since $\frac{1}{\sigma^*} = \sqrt{\frac{1}{\sigma_1^2} + \frac{1}{\sigma_2^2} + \cdots + \frac{1}{\sigma_T^2}}$ is fixed based on the final $\varepsilon, \delta$ we want to achieve (see

Equation 1), we need to balance various $\sigma_k$ accordingly. Given these observations, how much privacy budget we should spend for each iteration of our algorithm? A simple strategy is set same value of $\sigma_k$ for all $k$; this corresponds to spending the same privacy budget for extracting $k$-grams for all values of $k$.

One other heuristic is the following. In any data, we expect that 1-grams are more frequent than 2-grams, which should be more frequent than 3-grams and so on. We can afford to add larger noise in the beginning and add less noise in later stages. Thus one could set $\sigma_k = c\sigma_{k-1}$ for some $0 < c < 1$; that is, we decay the noise at a geometric rate. And thus we consume more privacy budget in extracting $k$-grams than $k-1$-grams. The effect of $c$ is shown in Figure 3. As expected, spending more privacy budget to extract longer $n$-grams produces more longer $n$-grams at the cost of smaller number of shorter $n$-grams. On the other hand, one can make a strategy to extract more shorter n-grams by setting $c > 1$.

**Pruning rule:** Figure 4 shows effect of using different pruning rules, $V_k = S_{k-1} \times S_1$ (single-side) or $V_k = (S_{k-1} \times S_1) \cap (S_1 \times S_{k-1})$ (both-side). While both-side pruning is a stricter rule compared to the single-side pruning and we expect to perform better overall, our experiments suggest that the situation is more intricate than our intuition. In particular, both rules to lead to similar number of discovered $n$-grams; However, it is interesting to see the distribution on the length of the output $n$-grams. The both-side pruning rule favors shorter length $n$-grams where as single-side pruning favors the longer length $n$-grams. We believe that understanding how pruning rules affect the performance of DPNE algorithms is an interesting research direction.

### 3.1.2 Comparison to $K$-anonymity and DPSU

$K$**-anonymity:** Figure 4 shows what fraction of $n$-grams whose count is at least $K$ is extracted by our algorithm, which is similar to comparing against $K$-anonymity based benchmarks. Figure 4 shows that DPNE algorithms can extract most of the $n$-grams which have frequency at least 100 and have length at least 5. In particular, our new algorithm could output $90\%$ of the $n$-grams of length 6,7,8 that appear in the dataset at least 100 times.

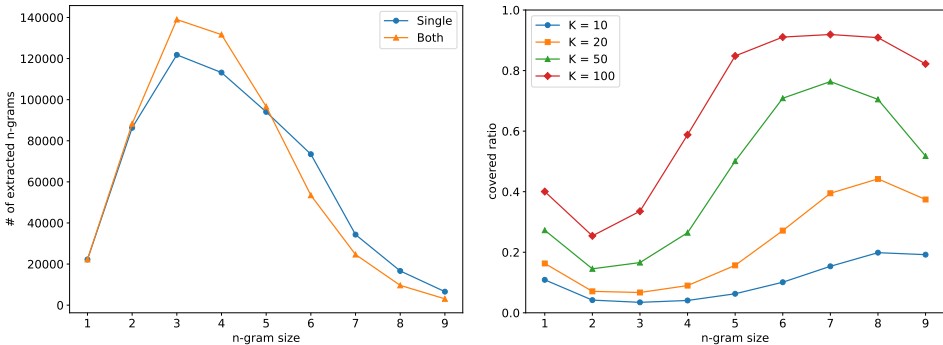

Figure 4: The left figure illustrates the effect of different pruning rules $V_k = S_{k-1} \times S_1$ (single-side) or $V_k = (S_{k-1} \times S_1) \cap (S_1 \times S_{k-1})$ (both-side). The right figure compares DPNE with $K$-anonymity, it shows how much fraction of $n$-grams with frequency $K$ are covered by the DPNE algorithm.

**DPSU:** In Table 4 and Figure 1, we compare our DPNE algorithm with the DPSU algorithm of [GGK$^+$20]. We implement DPSU algorithms to solve DPNE in three different ways:

- In DPSU-all, we run DPSU algorithm with the maximum user contribution $T\Delta_0$ and treat all $n$-grams similarly irrespective of their length.

- In DPSU-even, we allocate the privacy budget of $\approx \varepsilon/\sqrt{T}$ (using advanced composition) to extract the $n$-grams of each particular length $k$ separately, for each $k \in [1, 2, ..., 9]$. Maximum user contribution for each $k$ is set to $\Delta_0$.

- Finally, in DPSU-single, we allocate all the privacy budget of $\varepsilon$ towards extracting $n$-grams of a fixed length $k$, for each $k \in [1, 2, ..., 9]$. Maximum user contribution for each $k$ is set to $\Delta_0$.

Table 4: Comparison of DPNE and DPSU methods using parameters $\Delta_0 = 100, \varepsilon = 4, \delta = 10^{-7}, \eta = 0.01$. See Figure 1 for explanation of the different versions of DPSU.

|            | 1      | 2       | 3       | 4       | 5      | 6      | 7      | 8     | 9     | Total   |
|------------|--------|---------|---------|---------|--------|--------|--------|-------|-------|---------|
| DPNE       | 16,173 | 59,918  | 110,160 | 119,039 | 88,174 | 51,816 | 23,144 | 9,019 | 2,748 | 480,191 |
| DPSU-all   | 18,010 | 54,937  | 38,283  | 10,990  | 1,978  | 352    | 71     | 16    | 4     | 124,637 |
| DPSU-even  | 16,164 | 44,871  | 35,032  | 11,093  | 2,210  | 440    | 113    | 28    | 7     | 109,958 |
| DPSU-single| 44,635 | 194,131 | 201,901 | 92,486  | 25,451 | 5,906  | 1,385  | 329   | 92    | -       |

As we can see, except for smaller length $n$-grams, DPNE completely outperforms the DPSU algorithm. This is not surprising as DPSU algorithms do not exploit the rich structure inherent in $n$-grams. The most interesting statistic to note is the last row of Table 4 : Here we see that our new algorithm beats the DPSU algorithm even when all the privacy budget is allocated to extracting a certain fixed sized $n$-grams. For example, the DPSU algorithm with a privacy budget of $(\varepsilon = 4, \delta = 10^{-7})$ allocated to extracting *only* 8-grams managed to output only 329 8-grams, where as the DPNE algorithm with the same privacy budget spread across *all* the $n$-grams still could extract 9,019 8-grams. We also observe that the number of spurious $n$-grams output by our algorithm is always at most $\eta$-fraction of output as designed, so we omit the number of spurious $n$-grams from Table 4 for clarity.

### 3.2   Experiments on MSNBC Dataset

Table 5 reports comparison DPSU and DPNE algorithms on the MSNBC datasets. Although MSNBC dataset is significantly smaller than the Reddit dataset, behavior of the algorithms remain roughly the same.

Table 5: Comparison of DPSE and DPSU methods on MSNBC data ($\Delta_0 = 10, \varepsilon = 1, \delta = 10^{-7}, \eta = 0.01$)

|             | 1  | 2   | 3     | 4     | 5     | 6     | Total |
|-------------|----|-----|-------|-------|-------|-------|-------|
| DPNE        | 17 | 254 | 1,273 | 1,954 | 2,221 | 2,020 | 7,739 |
| DPSU-all    | 17 | 249 | 930   | 1,145 | 1,007 | 768   | 4,116 |
| DPSU-even   | 17 | 246 | 899   | 1,107 | 996   | 766   | 4,031 |
| DPSU-single | 17 | 260 | 1,483 | 2,180 | 2,210 | 1,819 | -     |

## 4   Conclusion

In this paper, motivated by its applications to NLP problems, we initiated the study of DPNE problem, and proposed new algorithms for the problem. We believe that our algorithmic framework can be extended or improved on multiple fronts: a more careful scheduling of privacy budget across different length $n$-grams, understanding the pruning strategies, etc. Given the ubiquitous nature of this problem in NLP, we hope that our work brings more attention to this problem.

## Acknowledgement

We would like to thank Robert Sim, Chris Quirk and Pankaj Gulhane for helpful discussions and encouraging us to work on this problem.

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
