# Differentially Private n-gram Extraction: Supplementary Materials

**Kunho Kim**
Microsoft
kuki@microsoft.com

**Sivakanth Gopi**
Microsoft Research
sigopi@microsoft.com

**Janardhan Kulkarni**
Microsoft Research
jakul@microsoft.com

**Sergey Yekhanin**
Microsoft Research
yekhanin@microsoft.com

## A   A scalable algorithm for DPNE

In this section, we will present a more scalable and faster version of Algorithm 1. The main observation is that we never actually need to explicitly calculate all the valid $k$-grams $V_k = (S_1 \times S_{k-1}) \cap (S_{k-1} \times S_1)$, since this set can be prohibitively big. Instead, we will use the fact that checking membership in $V_k$ is easy and we can also sample from $V_k$ relatively efficiently.

35th Conference on Neural Information Processing Systems (NeurIPS 2021), Sydney, Australia.

**Algorithm 1:** Algorithm for differentially private ngram extraction (Faster and Scalable Version)

---

**Input:** A set of $n$ users where each user $i$ has some subset $W_i^k$ of $k$-grams.
$T$: maximum length of ngrams to be extracted
$\Delta_1, \Delta_2, \ldots, \Delta_T$: maximum contribution parameters
$\rho_1, \rho_2, \ldots, \rho_T$: Threshold parameters
$\sigma_1, \sigma_2, \ldots, \sigma_T$: Noise parameters.
$p$: Sampling probability.
**Output:** $S_1, S_2, \ldots, S_T$ where $S_k$ is a set of $k$-grams
// Run DPSU to learn 1-grams
$S_1 \leftarrow$ Run Algorithm 5 (DPSU) using $\Delta_1, \rho_1, \sigma_1$ to get a set of 1-grams;
$V_1 \leftarrow S_1$;
// Iteratively learn $k$-grams
**for** $k = 2$ **to** $T$ **do**
    $\widetilde{\#V_k} \leftarrow$ EstimateValidKgrams$(S_1, S_{k-1}, p)$ ;           // Estimate valid $k$-grams
    Set $\rho_k$ using $|S_{k-1}|, \widetilde{\#V_k}$;
    // Build a weighted histogram using Weighted Gaussian policy
    $H_k \leftarrow$ Empty dictionary where any key which is inserted is initialized to 0;
    **for** $i = 1$ **to** $n$ **do**
        $U_i \leftarrow$ PruneInvalid$(W_i^k, S_1, S_{k-1})$ ;         // Prune invalid ngrams
        // Limit user contributions
        **if** $|U_i| > \Delta_k$ **then**
            $U_i \leftarrow$ Randomly choose $\Delta_k$ items from $U_i$;
        **for** $u$ *in* $U_i$ **do**
            $H_k[u] \leftarrow H_k[u] + \frac{1}{\sqrt{|U_i|}}$;

    // Add noise to $H_k$ and output $k$-grams which cross the threshold $\rho_k$
    $S_k = \{\}$ (empty set);
    **for** $u \in H_k$ **do**
        **if** $H_k[u] + N(0, \sigma_k^2) > \rho_k$ **then**
            $S_k \leftarrow S_k \cup \{u\}$;

    // Add spurious $k$-grams from $V_k \setminus \mathrm{supp}(H_k)$ with probability
       $\Pr[N(0, \sigma_k^2) > \rho_k] = \Phi(-\rho_k/\sigma_k)$
    $B_k = \mathsf{Binomial}(\widetilde{\#V_k} - |\mathrm{supp}(H_k)|, \Phi(-\rho_k/\sigma_k))$ ; // # Spurious $k$-grams we need to
    add to $S_k$
    $Sp_k \leftarrow \{\}$ ;                           // Spurious $k$-grams
    **while** $|Sp_k| < B_k$ **do**
        Sample random $x \sim S_1$ and $w \sim S_{k-1}$ uniformly and independently;
        Let $w = yz$ where $z \in S_1$;
        **if** $xy \in S_{k-1}$ *and* $z \in S_1$ *and* $w \notin (Sp_k \cup \mathrm{supp}(H_k))$ **then**
            $Sp_k \leftarrow w \cup Sp_k$;

    $S_k \leftarrow S_k \cup Sp_k$ ;      // Add the spurious $k$-grams to the $k$-grams extracted from
    users

Output $S_1, S_2, \ldots, S_T$;

---

**Algorithm 2:** EstimateValidKgrams: Algorithm for estimating number of valid $k$-grams

---

**Input:** $S_1$: Set of extracted 1-grams, $S_{k-1}$: Set of extracted $(k-1)$-grams, $p$: Sampling probability

**Output:** An estimate $\widetilde{\#V_k}$ for the number of valid $k$-grams $|V_k| = |(S_1 \times S_{k-1}) \cap (S_{k-1} \times S_1)|$

$N \leftarrow \lceil p|S_1||S_{k-1}| \rceil$;
$count \leftarrow 0$;
**for** $i = 1$ **to** $N$ **do**

    Sample random $x \sim S_1$ and $w \sim S_{k-1}$ uniformly and independently;
    Let $w = yz$ where $z \in S_1$;
    **if** $xy \in S_{k-1}$ *and* $z \in S_1$ **then**
        $count = count + 1$;

$\widetilde{\#V_k} \leftarrow \lceil count/p \rceil$;
Output $\widetilde{\#V_k}$;

---

**Algorithm 3:** CheckValidity: Check validity of a $k$-gram

---

**Input:** $w$: Any $k$-gram with $k \geq 2$, $S_1$: Set of extracted 1-grams, $S_{k-1}$: Set of extracted $(k-1)$-grams

**Output:** True if $w$ is valid i.e. $w \in V_k = (S_1 \times S_{k-1}) \cap (S_{k-1} \times S_1)$, else False

Let $w = xyz$ where $x, z$ are 1-grams;
**if** $x, z \in S_1$ *and* $xy, yz \in S_{k-1}$ **then**

    Output True;

**else**

    Output False;

---

**Algorithm 4:** PruneInvalid: Prune invalid $k$-grams from a given set of $k$-grams

---

**Input:** $W$: Any set of $k$-grams with $k \geq 2$, $S_1$: Set of extracted 1-grams, $S_{k-1}$: Set of extracted $(k-1)$-grams

**Output:** $W \cup V_k$ where $V_k = (S_1 \times S_{k-1}) \cap (S_{k-1} \times S_1)$

$\widehat{W} \leftarrow \{\}$;
**for** $w$ *in* $W$ **do**

    **if** *CheckValidity($w, S_1, S_{k-1}$)* **then**
        $\widehat{W} \leftarrow w \cup \widehat{W}$;

Output $\widehat{W}$;

---

# B  Differentially Private Set Union (DPSU) Algorithm

---

**Algorithm 5:** Algorithm for extracting 1-grams using DPSU

---

**Input:** A set of $n$ users where each user $i$ has some subset $W_i^1$ of 1-grams.
$\Delta_1$: maximum contribution parameter
$\rho_1$: Threshold parameter
$\sigma_1$: Noise parameter
**Output:** $S_1$, a set of 1-grams
**for** $i = 1$ *to* $n$ **do**
    $U_i \leftarrow W_i^1$;
    **if** $|U_i| > \Delta_1$ **then**
        $U_i \leftarrow$ Randomly choose $\Delta_1$ items from $W_i$;
    **for** $u$ *in* $U_i$ **do**
        $H_1[u] \leftarrow H_1[u] + \frac{1}{\sqrt{|U_i|}}$;

$S_1 = \{\}$ ;                                                                                    // `empty set`
$H_1 \leftarrow$ Empty dictionary where any key which is inserted is initialized to 0;
**for** $u \in H_1$ **do**
    **if** $H_1[u] + N(0, \sigma_1^2) > \rho_1$ **then**
        $S_1 \leftarrow S_1 \cup \{u\}$;

Output $S_1$;

---