# OpenReview forum: "Differentially Private n-gram Extraction"
_NeurIPS.cc/2021/Conference — NeurIPS 2021 Poster_

### Official Review · Reviewer_GGc1 · 2021-06-28

**Rating:** 7
**Confidence:** 4

**Summary:**

The paper provides an algorithm for n-gram extraction that is Differentially Private. They also provide a scalable version of the DP algorithm and provide experimental results.

**Main Review:**

The paper is well written and easy to follow. I also think the relevant literature is discussed properly. The main strength of this paper is the impact it can have on products as it tackles a real world problem where privacy is a concern. It is even discussed in the paper that the algorithm has been implemented in industry which I think is great. It could be argued that a weakness of this work is that the algorithm and its analysis follow very easily from previous work (in particular [GGK+20]). I personally do not consider the previous a weakness since I don’t think that important problems need complicated solutions. All the theoretical claims seem to be correct.

**Time Spent Reviewing:**

2.5

---

> ### Author Response · Authors · 2021-08-10
> **Thanks for your positive feedback.**
>
> Thanks for the positive feedback. We are happy to see that you agree with us that this is an important problem and deserves more research attention. We really appreciate it.

---

### Official Review · Reviewer_S4Sy · 2021-07-10

**Rating:** 7
**Confidence:** 3

**Summary:**

Paper studies the problem of n-gram extraction with differential privacy as a generalization of differentially private set union.

**Limitations And Societal Impact:**

Yes

**Main Review:**

paper proposes to deal with the problem of extracting n-grams with differential privacy posed as a generalization of DPSU. Paper is clearly written and deals with an important issue as n-gram extraction with privacy has multitude of potential applications as suggested by the authors. That being said, in some places (such as Section 2.1), it seems like some material is missing or has been left out, which is understandable due to page limits but can occasions make the reading a bit harder. My comments are provided below in detail:

1. Figure 1 shows the comparison for a fixed privacy budget, it will be nice to have similar comparisons with varying epsilon, even if they can go in supplementary material.
2. In contributions, last line stating that the proposed algorithms have been used in industry serve no purpose in my opinion.
3.  it will be helpful to add a line to state why is the construction of k-grams is a Gaussian mechanism with l_2 sensitivity=1
4. It will also help in the introduction, where authors state the dangers of leaking something like a SSN, to state how will a DP model solve the problem.
5. Minor: in the checklist, authors have answered "yes" to many questions that I cant seem to correlate with the paper, such as limitations, code, error bars, compute used, etc.

----------------------------------------------------------------------------------------------
post-rebuttal
I have read authors' response.


**Time Spent Reviewing:**

2.5

---

> ### Author Response · Authors · 2021-08-10
> **Thanks for the positive feedback.**
>
> Thanks for the positive review and great feedback. We appreciate it. Now we address your specific questions:
>
> 1.	More experiments on varying epsilon. Thanks for your suggestion. We did in fact do experiments with varying epsilon but didn’t get a chance to include it in the main body due to space limits. We will add it to the supplementary material as you suggested. Our algorithms do consistently well in the range of epsilons we tested.
>
> 2.	Comment on industry use. We totally agree with you that this comment serves no purpose from a scientific perspective. Our only motivation to include that line was to encourage more research on this problem as it is one of the widely used primitives in NLP applications. We believe that this problem deserves more attention from the DP community.
>
> 3.	Sensitivity of the Gaussian mechanism. Thanks for pointing this out, and we will explain why sensitivity is 1 in the next version.
>
> 4.	Discussion on how DP prevents leakage of private information. This is a nice catch, and in the next version, we will add a few lines of explanation on how DP can prevent such information leakage in the introduction.
>
> 5.	Checklist. Thanks for pointing this out. We will definitely go over our answers again and make sure to click the right answers. We apologize for any inconvenience.

---

### Official Review · Reviewer_Qhy6 · 2021-07-16

**Rating:** 6
**Confidence:** 4

**Summary:**

This paper proposes a differentially private algorithm to extract all n-grams of length 1 up to some maximum length T from a texture data corpus where each user’s data is represented by a single string. It uses an earlier algorithm called Differentially Private Set Union (DPSU) to find all 1-grams (the alphabet), then iteratively builds the set of n-grams of length from 2 to T by effectively pruning the search space, building a noisy weighted histogram on the pruned candidate set and taking the n-grams whose counts are above predetermined thresholds.

**Limitations And Societal Impact:**

As mentioned in the main review, the authors could expand their discussion on the limitations of their work. For example, how does the algorithm perform on dataset where the distribution of n-grams is highly concentrated? Can the algorithm be improved to use privacy budget more efficiently for those n-grams whose counts are very high and way above the threshold, as pointed out in the DPSU paper?

**Main Review:**

The problem of n-gram extraction is an important task in many NLP applications. In this paper, the authors propose a differentially private algorithm to extract all n-grams of length 1 up to some maximum length T from a texture data corpus where each user’s data is represented by a single string. Their algorithm is a simple extension of a previous algorithm called Differentially Private Set Union (DPSU), which aims to find all items owned by users where each user owns a set of items from some universe. The algorithm works as follows:
1.  Find S(1), the set of all 1-grams (the alphabet) using DPSU
2.  Iteratively build S(k), the set of k-grams for k from 2 to T. To build S(k) from S(k-1), it first prunes the search space for S(k) by restricting it to the intersection of S(1) \times S(k-1) and S(k-1) \times S(1). The rationale is that the two substrings of length (k-1) of a k-gram in S(k) should already be in S(k-1). Then it computes a noisy weighted histogram on the pruned search space and extracts k-grams whose counts are above predefined thresholds, which are considered hyperparameters.

The novel part is the combination of DPSU with effective pruning. DPSU does not consider the structures of items and will regard all n-grams as unrelated entities. When combined with tree-based pruning approach, it will produce improved results on extraction of n-grams, as is verified by the experiment.


Strength:
+ The problem formation is clear. The authors clearly state what problem they are trying to address.

+ The algorithm is simple and intuitive, and it works! The improved performance is demonstrated by experiments.

+ The paper is well-written. The presentation is clear.

Weakness:
- In all experiments the authors report the number of n-grams extracted or the coverage ratio. However, it is also important to know the number of false positives, or the so-called spurious n-grams in the paper. Thus, it would be more desirable for the experiments to also report other metrics such as the F-scores.

- Limitations of the algorithm are not adequately discussed. The algorithm relies on the set of shorter n-grams to build the set of longer n-grams. Any error that happens in the early stage will propagate to subsequent computations. The paper argues that for privacy budget allocation, one could use smaller budgets and thus larger noises for shorter n-grams because shorter n-grams are more frequent than longer ones and have larger counts. However, there is no analysis or experiment to back up this claim and I am not convinced this is true. After all, the algorithm is to identify as many n-grams that appear in the corpus, not just the frequent ones.

- The part on related work could be expanded. There are many related algorithms that aim to identify existing n-grams, both in the central DP setting and the local DP setting. Those works should be mentioned to provide a more thorough review of the literature in this area.

Typos:
line 138, bother -> both


**Time Spent Reviewing:**

6

---

> ### Author Response · Authors · 2021-08-10
> **Thanks for the detailed feedback.**
>
> Thank you for positive and detailed feedback. We appreciate it.  Now we address your specific questions:
>
> 1.	Experiments on false-positive or spurious n-grams.   We did not plot the number of spurious n-grams output by our algorithm experimentally because Proposition 2.2 gives a precise bound on this quantity, and we have also experimentally verified that this bound holds, i.e., the spurious n-grams are always at most an $\eta$ fraction of the output where $\eta$ is a hyperparameter which we set to $0.01$ in our experiments. Figure 3 (left) shows the effect of $\eta$ on the total output. Given an opportunity, we will plot the number of spurious n-grams output by our algorithm and experimentally confirm that they are indeed at most an $\eta$-fraction of the output.
>
> 2.	Privacy budget allocation across different n-grams:  We agree with you that our suggestion on using a smaller privacy budget for shorter n-grams and allocating larger privacy budget for longer n-grams is based on a heuristic argument.  Figure 3 (right) shows one such experiment supporting our intuition, by giving more privacy budget to discover longer n-grams (which is often desired in practice), we can discover more longer n-grams. This is achieved by setting the hyperparameter $c<1$. Qualitatively, this experiment supports our intuition. If $c<1$, we will find more longer n-grams and if $c>1$ we will find more shorter n-grams.  In summary, the hyperparameter $c$ gives a way to control the distribution of lengths of n-grams output by our algorithm. Unfortunately, from the figure it is difficult to see how the total number of n-grams varies as $c$ changes. We will make this clear in the next version.
>
> 3.	Regarding allocating less budget to frequent n-grams: This is a great point, and exactly the reason DPSU algorithms (Gopi et al) outperform histogram based algorithms for 1-gram extraction problem.  Note, however, that our algorithm for DPNE can implicitly exploit this idea as well, because each iteration of our algorithm can use any DPSU policy to extract as many k-grams as possible for k= 1, 2….n. Thus, implicitly, DPNE also spends less privacy budget on extracting frequent n-grams. In our paper, we only use the weighted-gaussian DPSU update policy to learn k-grams, this is because it scales well for very large datasets. But in principle, we can use any DPSU policy, we will make this clear in a future revision. It is possible that there are better strategies to exploit this fact than our proposed framework. We believe that it is a great research direction given the ubiquitous nature of this problem in NLP. We will expand on this discussion in the next version.
>
> 4.	Expanding related work. Thanks for your feedback and we will expand on the works we have cited. We were concise in our discussion of the related work due to page limits and also because we did not find many papers (except DPSU) that tackled exactly the n-gram extraction problem we are studying. We would be grateful if you can point out the papers we have missed and should be cited.
>
> Thanks also for pointing out the typos and other comments. We will incorporate them in the next version.

---

### Official Review · Reviewer_mTUU · 2021-07-20

**Rating:** 6
**Confidence:** 3

**Summary:**

This paper studies the problem of differentially private n-gram extraction (DPNE). The authors combines the idea of DPSU and a tree based approach to solve the DPNE problem, where the basic building blocks are histogram construction and the Gaussian mechanism.

**Limitations And Societal Impact:**

See above.

**Main Review:**

- DPNE has wide applications, but there has not been much research on this problem in the setting of differential privacy.
- The experiments show that the proposed method practically outperforms the DPSU on DPNE problem.
- The theoretical depth is weak: there is no clear accuracy guarantee and the privacy analysis is straightforward.

**Time Spent Reviewing:**

1

---

> ### Author Response · Authors · 2021-08-10
> **Thanks for the positive feedback**
>
> Thanks for your positive review. We do hope that our work spurs more research on this problem, as this is an important subroutine in many NLP applications based on our experience.
>
> About the lack of provable utility guarantees: To prove such guarantees, we would need to make some distributional assumptions on the input dataset. This is because in the worst case utility analysis one can show strong lower bounds even for DPSU.  Since any simplistic distributional assumptions on the inputs would be far from real world datasets, we felt that it wouldn’t be of much practical utility. Therefore, we prove the privacy bounds rigorously and to demonstrate that it has good utility, we rely on experimental evaluation on real world datasets.

---

### Decision · Program_Chairs · 2021-09-27

**Decision:**

Accept (Poster)

**Comment:**

This paper proposes a DP algorithm to extract all n-grams (sequence of n consecutive words appearing in the data) of length 1 up to some maximum length T, where each user’s data is represented by a single string. It uses an earlier algorithm to find all 1-grams, then iteratively builds the set of n-grams of length from 2 to T by effectively pruning the search space. The reviewers agree that this is an interesting paper, and all of them support acceptance.